# Occupation and SARS-CoV-2 seroprevalence studies: a systematic review

Emily Boucher [1], Christian Cao,[1] Sean D'Mello,[2] Nathan Duarte,[3] Claire Donnici,[1] Natalie Duarte,[4] Graham Bennett,[5] SeroTracker Consortium, Anil Adisesh [6,7,8] Rahul Arora,[1,9] David Kodama,[6,10] Niklas Bobrovitz [11,12]

## ABSTRACT

**Objective** To describe and synthesise studies of SARS-CoV-2 seroprevalence by occupation prior to the widespread vaccine roll-out.

**Methods** We identified studies of occupational seroprevalence from a living systematic review (PROSPERO CRD42020183634). Electronic databases, grey literature and news media were searched for studies published during January–December 2020. Seroprevalence estimates and a free-text description of the occupation were extracted and classified according to the Standard Occupational Classification (SOC) 2010 system using a machine-learning algorithm. Due to heterogeneity, results were synthesised narratively.

**Results** We identified 196 studies including 591 940 participants from 38 countries. Most studies (n=162; 83%) were conducted locally versus regionally or nationally. Sample sizes were generally small (median=220 participants per occupation) and 135 studies (69%) were at a high risk of bias. One or more estimates were available for 21/23 major SOC occupation groups, but over half of the estimates identified (n=359/600) were for healthcare-related occupations. 'Personal Care and Service Occupations' (median 22% (IQR 9–28%); n=14) had the highest median seroprevalence.

**Conclusions** Many seroprevalence studies covering a broad range of occupations were published in the first year of the pandemic. Results suggest considerable differences in seroprevalence between occupations, although few large, high-quality studies were done. Well-designed studies are required to improve our understanding of the occupational risk of SARS-CoV-2 and should be considered as an element of pandemic preparedness for future respiratory pathogens.

## STRENGTHS AND LIMITATIONS OF THIS STUDY

⇒ We conducted a comprehensive search of the COVID-19 seroprevalence literature, including non-English articles, government reports, unpublished data.

⇒ Occupations were classified using the Standard Occupational Classification 2010 coding system to improve interpretability and facilitate comparison with other datasets.

⇒ Seroprevalence may underestimate the true prevalence of infection because antibody titres decline over time, but where possible we prioritised prevalence estimates for IgG antibodies, which appear to be more robust than other immunoglobulin types.

⇒ We did not adjust for differences in serological test performance.

For numbered affiliations see end of article.

**Correspondence to**
Emily Boucher;
emily.boucher@ucalgary.ca

## INTRODUCTION

Occupation is a social determinant of health and an important risk factor for SARS-CoV-2 infection. Essential workers in health and social care occupations have an increased risk of COVID-19 compared with non-essential workers, but the risks for other occupations are not well defined.[1–3] Studies using confirmed COVID-19 cases to examine occupational COVID-19 risk are affected by variable testing rates. For example, testing rates may be higher in workplaces offering testing or paid sick leave, and are impacted by geographic (eg, urban vs rural) and socioeconomic factors (eg, deprivation), potentially biasing results.[4–6] Few high-quality, prospective studies using frequent, serial molecular or antigen testing covering a broad range of occupations have been conducted, in part due to the costs and administrative burden of such studies.[7 8]

Serological testing for SARS-CoV-2 antibodies provides evidence of previous infection and/or vaccination depending on vaccination status and the specific antigens targeted and can be used to obtain more accurate estimates of the cumulative incidence of infection.[9] Accurate data on the occupational risks of COVID-19 and other respiratory infections are essential for informing the development of occupational safety guidelines and regulations, transmission control measures and resource allocation (testing, personal protective equipment (PPE), etc). The objectives of this review were to describe and synthesise studies of SARS-CoV-2 seroprevalence across a broad range of occupations globally prior to the widespread roll-out of vaccines.

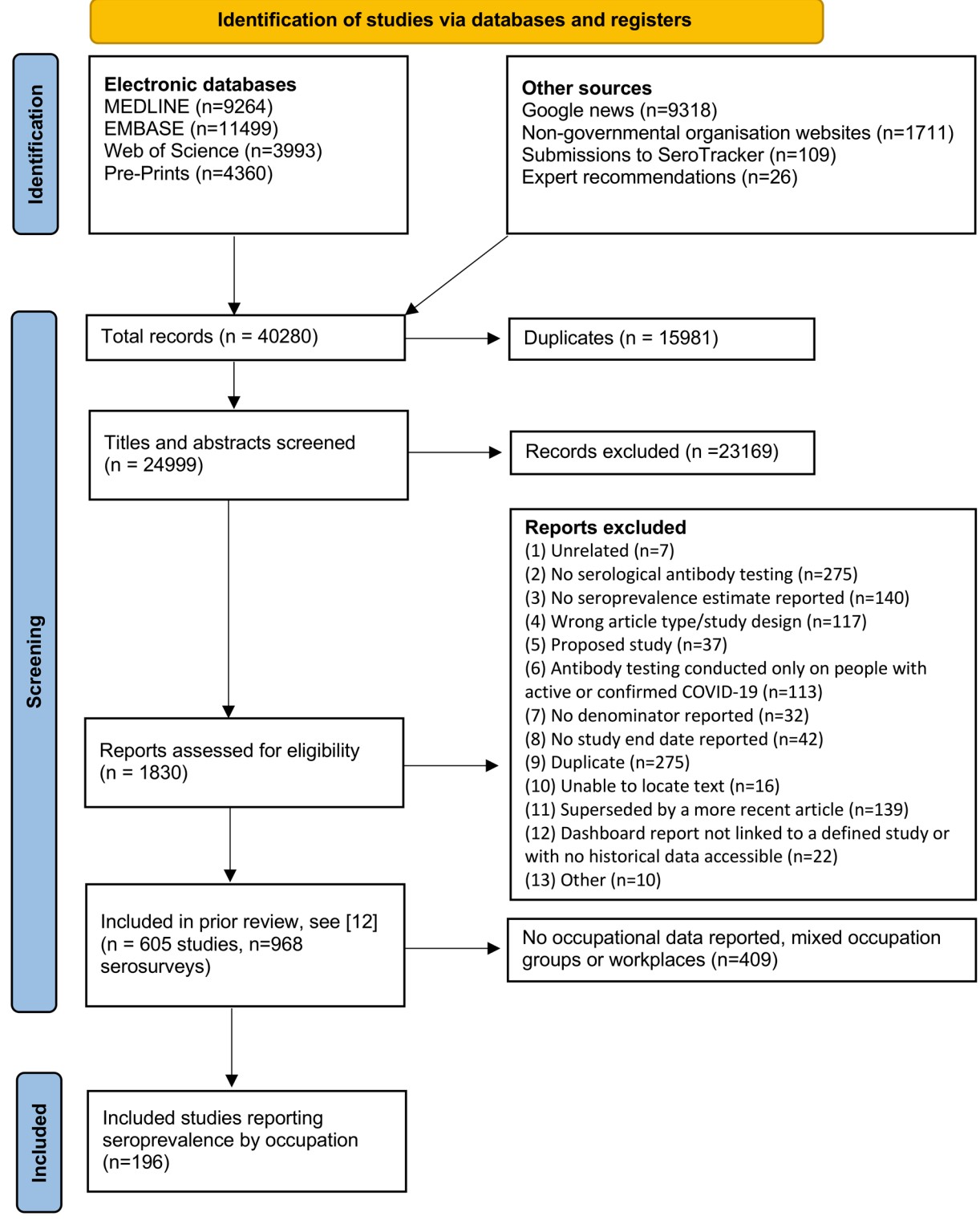

*From:* Page MJ, McKenzie JE, Bossuyt PM, Boutron I, Hoffmann TC, Mulrow CD, et al. The PRISMA 2020 statement: an updated guideline for reporting systematic reviews. BMJ 2021;372:n71. doi: 10.1136/bmj.n71

For more information, visit: http://www.prisma-statement.org/

**Figure 1** PRISMA flow diagram, Page *et al*.[18] PRISMA, Preferred Reporting Items for Systematic Reviews and Meta-Analyses.

## METHODS

We identified seroprevalence studies with sample frames or subgrouping variables related to occupation or employment status from a database compiled via a living systematic review (PROSPERO CRD42020183634). The database has been described previously and includes

| SOC 2010 Major Occupation Group | Total | | Median, IQR | | Seroprevalence % | | N, % |
|---|---|---|---|---|---|---|---|
| | Estimates | Countries | Study dates, midpoint | Sample size | (Median, IQR) | (Scale 0-75%) | Low-Moderate RoB |
| Architecture and Engineering Occupations (17-0000) | 1 | 1 | 15/08 (15/08-15/08) | 21 (21-21) | 42.9 (42.9-42.9) | | 0 (0%) |
| Personal Care and Service Occupations (39-0000) | 14 | 7 | 03/05 (02/04-02/06) | 127 (54-302) | 21.5 (9.32-27.76) | | 3 (21%) |
| Installation, Maintenance, and Repair Occupations (49-0000) | 1 | 1 | 19/06 (19/06-19/06) | 134 (134-134) | 16.4 (16.4-16.4) | | 0 (0%) |
| Building and Grounds Cleaning and Maintenance Occupations (37-0000) | 17 | 8 | 13/07 (09/06-16/08) | 102 (42-226) | 10.8 (3.3-21.7) | | 6 (35%) |
| Healthcare Support Occupations (31-0000) | 39 | 12 | 05/06 (19/05-21/06) | 263 (122-562) | 10.7 (2-20.05) | | 12 (31%) |
| Business and Financial Operations Occupations (13-0000) | 2 | 2 | 05/07 (18/06-22/07) | 462 (252-671) | 8.27 (5.3-11.23) | | 2 (100%) |
| Management Occupations (11-0000) | 10 | 6 | 17/06 (01/05-02/08) | 44 (23-145) | 8.17 (6.7-19.93) | | 3 (30%) |
| Food Preparation and Serving Related Occupations (35-0000) | 6 | 4 | 17/06 (11/05-23/07) | 58 (12-108) | 6.35 (2.37-24.03) | | 2 (33%) |
| Healthcare Practitioners and Technical Occupations (29-0000) | 222 | 23 | 13/06 (13/05-13/07) | 215 (64-482) | 5.91 (1.83-11.71) | | 84 (38%) |
| _Healthcare Practitioners and Technical Occupations, 5-digit codes**_ | | | | | | | |
| Miscellaneous Health Technologists and Technicians | 4 | 3 | 26/08 (09/08-12/09) | 60 (20-121) | 12.96 (9.09-27.54) | | 1 (25%) |
| Registered Nurses | 78 | 18 | 05/06 (05/05-05/07) | 329 (71-1000) | 8.44 (3.68-15.5) | | 22 (28%) |
| Clinical Laboratory Technologists and Technicians | 18 | 12 | 15/06 (19/05-11/07) | 204 (86-284) | 6.22 (2.07-11.94) | | 12 (67%) |
| Physicians and Surgeons | 65 | 21 | 09/06 (10/05-09/07) | 214 (59-564) | 5.88 (1.85-11.8) | | 23 (35%) |
| Emergency Medical Technicians and Paramedics | 9 | 6 | 13/06 (27/05-30/06) | 157 (56-243) | 5.41 (5.2-11) | | 4 (44%) |
| Therapists | 15 | 4 | 08/06 (19/05-28/06) | 121 (61-235) | 3.75 (0-9.45) | | 7 (47%) |
| Physician Assistants | 9 | 2 | 27/06 (26/05-28/07) | 230 (156-320) | 3.48 (0.64-9.43) | | 3 (33%) |
| Pharmacists | 9 | 7 | 29/06 (14/06-14/07) | 113 (29-213) | 0.5 (0-3.45) | | 4 (44%) |
| Healthcare Occupations (mixed)* | 94 | 25 | 05/06 (29/04-12/07) | 375 (110-1012) | 5.66 (2.35-11.6) | | 23 (24%) |
| Sales and Related Occupations (41-0000) | 23 | 8 | 21/08 (22/06-19/10) | 643 (236-1184) | 5.3 (1.2-8.8) | | 6 (26%) |
| Education, Training, and Library Occupations (25-0000) | 6 | 5 | 05/07 (12/06-27/07) | 238 (73-1305) | 5.07 (2.71-17.22) | | 3 (50%) |
| Farming, Fishing, and Forestry Occupations (45-0000) | 3 | 3 | 13/07 (25/06-30/07) | 80 (66-100) | 5 (2.5-5) | | 1 (33%) |
| Not employed (mixed)* | 37 | 14 | 23/06 (12/05-04/08) | 382 (116-905) | 4.9 (2.7-14.97) | | 28 (76%) |
| Office and Administrative Support Occupations (43-0000) | 39 | 18 | 14/06 (18/05-11/07) | 120 (32-522) | 4.88 (1.36-13.36) | | 20 (51%) |
| First responders (mixed)* | 6 | 1 | 18/05 (13/05-22/05) | 219 (72-599) | 4.67 (1.6-7.34) | | 1 (17%) |
| Community and Social Service Occupations (21-0000) | 6 | 2 | 30/05 (18/05-11/06) | 104 (49-188) | 4.45 (2.13-6.1) | | 1 (17%) |
| Protective Service Occupations (33-0000) | 28 | 9 | 04/07 (21/05-16/08) | 190 (46-555) | 4.29 (2.17-7.47) | | 6 (21%) |
| Transportation and Material Moving Occupations (53-0000) | 23 | 7 | 08/08 (08/06-08/10) | 230 (80-364) | 3.5 (1.8-11.8) | | 8 (35%) |
| Life, Physical, and Social Science Occupations (19-0000) | 11 | 7 | 06/07 (11/06-30/07) | 343 (174-570) | 2.6 (1.66-6.46) | | 4 (36%) |
| Production Occupations (51-0000) | 4 | 3 | 23/05 (26/04-19/06) | 764 (342-1132) | 1.52 (1.45-4.93) | | 2 (50%) |
| Arts, Design, Entertainment, Sports, and Media Occupations (27-0000) | 6 | 5 | 07/07 (04/06-09/08) | 164 (47-823) | 1.39 (0.18-11.02) | | 3 (50%) |
| Computer and Mathematical Occupations (15-0000) | 1 | 1 | 03/05 (03/05-03/05) | 47 (47-47) | 0 (0-0) | | 1 (100%) |
| Construction and Extraction Occupations (47-0000) | 1 | 1 | 03/05 (03/05-03/05) | 42 (42-42) | 0 (0-0) | | 1 (100%) |

**Figure 2** Seroprevalence by SOC 2010 major occupation group. *Estimates are a mix of 'Healthcare Practitioners and Technical Occupations' and 'Healthcare Support Occupations'. SOC, Standard Occupational Classification.

>1000 cohort and cross-sectional studies reporting antibody testing for SARS-CoV-2 in humans identified from electronic databases, grey literature and news media.[10–12] We restricted the current review to studies published during January–December 2020 before vaccines were rolled-out, because differential vaccination rates by occupation may obscure results. We excluded studies that only reported seroprevalence for mixed occupation groups or workplaces (eg, 'hospital staff') rather than specific occupations, included children <18 years and that could not be machine-translated using Google Translate if unavailable in English or French (online supplemental file 1).

We extracted study information, sample characteristics, seroprevalence estimates and study-level risk of bias from the living review database. Risk of bias was assessed with a modified Joanna Briggs Institute Checklist for Prevalence Studies by one reviewer and verified independently as described previously. Overall risk of bias was assessed qualitatively based on whether seroprevalence estimates were very likely (corresponding to a low risk of bias), likely (moderate risk) or unlikely (low risk) to be correct

for the author's stated target population (online supplemental file 1).[12 13] If multiple estimates were reported, the most recent estimate using laboratory-based methods (eg, ELISA) and anti-spike and/or IgG antibodies were prioritised, because non-IgG and anti-nucleocapsid antibodies may decline more rapidly.[14] Free-text descriptions of occupations were extracted from the original studies by one researcher and reviewed by a second.

For each seroprevalence estimate, we identified the relevant Standard Occupational Classification (SOC) 2010 codes by applying the National Institute for Occupational Safety and National Institute for Health Industry and Occupation Computerised Coding System (NIOCCS) to occupation descriptions.[15] NIOCCS was chosen, because many studies were conducted in the USA. Coding was manually verified if there was insufficient information for NIOCCS classification, or if the probability of correct classification to the six-digit level was <0.8 based on our review of a subset of the NIOCCS coded data (online supplemental file 1). Anticipating substantial heterogeneity and an insufficient number of estimates relative to

covariates for meta-regression, we planned to summarise data using the median/IQR.

## Patient and public involvement
It was not possible or appropriate to involve patients or the public in this study.

## RESULTS
We identified 196 studies of occupational seroprevalence conducted in 2020 during the first and second waves of the pandemic (figure 1). There were 591 940 participants from 38 countries, including the USA (n=44 studies), UK (n=16) and Italy (n=15). Most studies (n=162; 83%) were conducted locally (eg, city, county) as opposed to regionally (eg, state; n=20; 10%) or nationally (n=14; 7%). Most were restricted to one occupational group (n=103), limiting direct comparisons (ie, using the same reference group). Sample sizes were often small (median=220, IQR 64–568 participants). Overall, 135 studies (69%) were at a high risk of bias, 47 moderate (24%), 2 low (1%) and 12 unclear (6%). Common reasons for bias were inadequate statistical analysis (ie, no adjustment for test or sample characteristics; 92%), non-probability sampling (74%) and small sample size (46%).

At least one estimate was available for all 23 major SOC occupation groups, except for 'legal' and 'military-specific' occupations (figure 2; all studies). Over half of the 600 estimates identified (n=359) were for healthcare-related occupations. For SOC groups with three or more estimates, the highest median seroprevalence was reported for 'personal care and service occupations' (median 22% (IQR 9%–28%); n=14, eg, 'personal care aids'). The next highest was reported for 'building and grounds cleaning and maintenance' occupations (11% (3%–22%); n=17, for example, 'maids and housekeeping cleaners') and 'healthcare support' (11% (2%–20%); n=39, eg, 'nursing assistants') occupations. The lowest median seroprevalence was 1% (0%–11%; n=6, eg, 'athletes') for 'arts, design, entertainment, sports and media occupations.' Individual estimates are listed in online supplemental file 2.

## DISCUSSION
This review is the first comprehensive synthesis of occupational COVID-19 seroprevalence studies worldwide. We identified 196 studies representing 21 out of 23 major SOC groups conducted during the first and second waves of the SARS-CoV-2 pandemic in 2020, prior to the widespread roll-out of vaccines, and described occupational groups with high seroprevalence.

Seroprevalence studies may estimate the cumulative incidence of infection more accurately than diagnostic testing studies when access to testing and test performance are poor, and also can identify asymptomatic infections.[6][8] The data identified suggest considerable differences in seroprevalence by occupation, though we did not statistically test for differences due to considerable variation in geography, study dates and workplace determinants of infection (eg, PPE, ventilation). 'Caring and personal service' occupations had the highest median seroprevalence (22%), which was four times higher than the unemployed (5%) and median seroprevalence across all occupational groups (5%). The UK Office for National Statistics reported a slightly lower cumulative incidence for positive diagnostic or rapid tests for COVID-19 across 25 occupational groups of 4% (mean),[4] but the discrepancy between the true cumulative incidence and confirmed infections is likely greater in regions with less access to testing: some national, population-based sero-surveys have estimated there are 10–20 serologically identifiable cases per 1 confirmed case.[12]

In future pandemics, large, well-reported, high-quality seroprevalence studies across a broad range of occupations are needed at an early stage to inform appropriate workplace policy. It has been suggested that 20% of the US workforce was exposed to disease or infection at work at least once a month prior to the pandemic.[16] Accurate data on the occupational risks of respiratory infections, including SARS-CoV-2, are needed to inform understanding of transmission, occupational health and safety agency guidelines and allocation of resources (eg, PPE and vaccines) during outbreaks and pandemics. For governments, there are also issues of occupational disease recognition and compensation to be considered.

As such, future population-based studies on respiratory infections should collect data on occupation. In the case of epidemic infection, collaboration between academic centres with the capacity to conduct large-scale studies and government agencies with expertise in disease surveillance and access to workplace data (eg, public health, occupational health and safety) may be beneficial.[12] Other authors have suggested the utility of occupational surveillance systems.[17] However, the routine completion of the occupation field in electronic health records would also serve this purpose as well as informing patient reported outcome measures.

### Strengths and limitations
Despite the large number of studies of occupational seroprevalence conducted, many studies had methodological limitations. Only two studies were at a low risk of bias and most occupational subgroups had small sample sizes (median 220 participants). Many were limited to one major SOC group (n=103 studies), which precluded comparisons. Detailed descriptions of occupations were often lacking, potentially contributing to coding errors and misclassification, and workplace determinants of infection (eg, use of PPE) were poorly reported.

In conclusion, our review shows that a large number of seroprevalence studies covering a broad range of occupations were published in the first year of the pandemic. Results suggest considerable differences in seroprevalence between occupations, although few large, well-reported, high-quality studies were done. Carefully

designed, adequately powered seroprevalence studies with coverage of a broad range of occupations could improve our understanding of the occupational risk of SARS-CoV-2 and other respiratory infections and should be considered an element of pandemic preparedness and response.

**Author affiliations**
[1]Cumming School of Medicine, University of Calgary, Calgary, Alberta, Canada
[2]Faculty of Engineering, University of Waterloo, Waterloo, Ontario, Canada
[3]Faculty of Engineering, McGill University, Montreal, Québec, Canada
[4]Faculty of Arts and Science, University of Toronto, Toronto, Ontario, Canada
[5]Department of Economics, McGill University, Montreal, Québec, Canada
[6]St. Michael's Hospital, Unity Health Toronto, Toronto, Ontario, Canada
[7]Division of Occupational Medicine, University of Toronto, Toronto, Ontario, Canada
[8]Canadian Health Solutions, Saint John, New Brunswick, Canada
[9]Institute of Biomedical Engineering, Oxford University, Oxford, UK
[10]Division of Emergency Medicine, University of Toronto Department of Medicine, Toronto, Ontario, Canada
[11]Temerty Faculty of Medicine, University of Toronto, Toronto, Ontario, Canada
[12]Department of Critical Care Medicine, University of Calgary, Calgary, Alberta, Canada

**Collaborators** SeroTracker Consortium: Cheng Matthew P, Donnici Claire, Illincic Natasha, Liu Michael, Papenburg Jesse, Segal Mitchell J, Penny Lucas J, Perlman-Arrow Sara, Rahim Hannah P, Yan Tingting, Yanes-Lane Mercedes.

**Contributors** This secondary analysis of the SeroTracker database was conceived by NB, EB, DK and AA. Senior authors on this paper were NB, DK, RA and AA. The protocol was developed by EB, NB and DK. Data cleaning was performed by CC, CD, NataD, SD'M and EB and verification by EB, SD, NathD and GB. Analysis was performed by EB and RA. The first draft of the manuscript was written by EB and revised by EB, RA, NB, NathD, GB, S'M, CC, AA, DK. The SeroTracker Consortium maintained the living systematic review database used in the study. All authors reviewed and agreed to the findings, and also provided critical revisions to the paper. EB accepts full responsibility for the work and/or the conduct of the study, had access to the data, and controlled the decision to publish.

**Funding** SeroTracker receives funding for SARS-CoV-2 seroprevalence study evidence synthesis from the Public Health Agency of Canada through Canada's COVID-19 Immunity Task Force (Grant Number 2021-HQ-000056), the WHO Health Emergencies Programme, the Robert Koch Institute and the Canadian Medical Association Joule Innovation Fund.

**Disclaimer** No funding source had any role in the design of this study, its execution, analyses, interpretation of the data, or decision to submit results. This manuscript does not necessarily reflect the views of the WHO or any other funder.

**Competing interests** RA was previously a Technical Consultant for the Bill and Melinda Gates Foundation Strategic Investment Fund, is a minority shareholder of Alethea Medical and was a former Senior Policy Advisor at Health Canada. Each of these relationships is unrelated to the present work. JP reports grants to his institution from MedImmune, Sanofi Pasteur, Merck and AbbVie, and personal fees for lectures from AbbVie and Astra-Zeneca, all outside of the submitted work. MPC reports grants from McGill Interdisciplinary Initiative in Infection and Immunity, grants from Canadian Institutes of Health Research, during the conduct of the study; personal fees from GEn1E Lifesciences, personal fees from nplex biosciences, personal fees from Kanvas biosciences, personal fees from AstraZeneca, non-financial support from Cidara therapeutics, non-financial support from Scynexis, non-financial support from Amplyx Pharmaceutics, outside the submitted work. In addition, MPC has a patent for methods detecting tissue damage, graft versus host disease, and infections using cell-free DNA profiling pending, a patent for methods assessing the severity and progression of SARS-CoV-2 infections using cell-free DNA pending, a patent for rapid identification of antimicrobial resistance and other microbial phenotypes using highly multiplexed fluorescence in situ hybridisation pending, and a patent highly multiplexed detection of gene expression with hybridisation chain reaction pending, all outside the submitted work.

**Patient and public involvement** Patients and/or the public were not involved in the design, or conduct, or reporting, or dissemination plans of this research.

**Patient consent for publication** Not applicable.

**Provenance and peer review** Not commissioned; externally peer reviewed.

**Data availability statement** SeroTracker data are available in a public, open access repository. All data relevant to the study are included in the article or uploaded as online supplemental information. Seroprevalence data can be downloaded (or requested) from https://serotracker.com.

**ORCID iDs**
Emily Boucher http://orcid.org/0000-0002-9854-3462
Anil Adisesh http://orcid.org/0000-0002-4973-8474
Niklas Bobrovitz http://orcid.org/0000-0001-7883-4484

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
