## [Reviewer comments · BMJ Open]

ARTICLE DETAILS

TITLE (PROVISIONAL)	Occupation and SARS-CoV-2 seroprevalence studies: a systematic review
AUTHORS	Boucher, Emily; Cao, Christian; D'Mello, Sean; Duarte, Nathan; Donnici, Claire; Duarte, Natalie; Bennett, Graham; Consortium, SeroTracker; Adisesh, Anil; Arora, Rahul; Kodama, David; Bobrovitz, Niklas

VERSION 1 – REVIEW

REVIEWER	Melissa Sutton Oregon Health Authority, Acute & Communicable Disease Prevention
REVIEW RETURNED	20-Jun-2022

GENERAL COMMENTS	PAGE 5  -Line 18: I would clarify variable testing rates and access. The categories in parentheses probably warrant additional comment. -Lines 18-22: 'serial diagnostic' is a bit of a misnomer. Serial testing of individuals without symptoms or exposure is screening testing. Diagnostic testing refers to testing of individuals with symptoms or exposure. Do you want to expand on why these haven't been done, e.g., cost prohibitive? -Lines 36-38: I would call out occupational safety guideline creation. PAGE 6  -Line 6: machine translatable using which software? -Line 38: I'm unfamiliar with this probability classification cut-off. Will the reader understand without additional information? PAGE 8  -Line 20-23: I think there are additional issues beyond testing access, e.g., test characteristics, the proportion of asymptomatic infection. -Line 34-39: I'm not following the 4% statistic--what is a mean risk of a positive test? Are you referring to percent positivity or a measurement of cumulative incidence at some point in the pandemic? PAGE 9  -Lines 6-8: "to inform compliance with workplace safety regulations" is not really the domain of public health. I think the point here is that we need occupational risk data to inform occupational health and safety agency guidelines. And, on the public health side, we need to use these data to inform PPE and vaccine allocation. -Lines 15-19: hmmm. Public health has very real constraints and must prioritize work especially during an early pandemic. I'm not sure this is a realistic recommendation. I'm also not totally clear on why this is being recommended? -Lines 35-36: contributing to misclassification? PAGE 10:
--

	-Line 6: pandemic preparedness or response? Overall comments: -Why no traditional Figure 1? -Very limited number of references, strongly suggest expanding literature review to inform manuscript. -PRISMA check list?
--	---

REVIEWER	Chao Wu Nanjing University
REVIEW RETURNED	28-Jun-2022

GENERAL COMMENTS	Boucher et al described and analyzed studies of SARS-CoV-2 seroprevalence by occupation prior to the widespread vaccine rollout, which seroprevalence studies covering a broad range of occupations were published in the first year of the pandemic. Results suggest considerable differences in seroprevalence between occupations, only few large, high-quality studies were done. Well designed studies are required to improve our understanding of the occupational risk of SARSCoV-2 and should be considered as an element of pandemic preparedness for future respiratory pathogens. However, after a careful reading of the article, I think that the importance of this review is not reflected, and the important conclusions of this study are not well presented. The major issue are present study only described a broad range of occupations of seroprevalence in first years of pandemic, The epidemic situation, prevention and control policies, economy and working methods, as well as the behavior and habits of the people in each region were not analyzed and elaborated. Because the disease that spreads in a respiratory tract is in epidemic process, individual life habit and protection consciousness, economic status is the most important factor that causes epidemic, the characteristic of occupation is not obvious. Therefore, the description and analysis of the medical service industry, that is, the relevant research on these places of occupational exposure risk is more important.
--

VERSION 1 – AUTHOR RESPONSE

Reviewer: 1

Dr. Melissa Sutton, Oregon Health Authority

Comments to the Author:

PAGE 5

Line 18: I would clarify variable testing rates and access. The categories in parentheses probably warrant additional comment.

Reponses to reviewer: We appreciate the reviewer’s comment and have clarified and expanded on this sentence. We now state that, “Studies examining confirmed COVID-19 cases to examine occupational COVID-19 risk are affected by variable testing rates. For example, testing rates may be higher in workplaces offering testing or paid sick leave, and are impacted by geographic (e.g., urban

versus rural) and socio-economic factors (e.g., deprivation), potentially biasing results” (Page 4, Lines 17-20).

Lines 18-22: 'serial diagnostic' is a bit of a misnomer. Serial testing of individuals without symptoms or exposure is screening testing. Diagnostic testing refers to testing of individuals with symptoms or exposure. Do you want to expand on why these haven't been done, e.g., cost prohibitive?

Responses to reviewer: We have changed the text to specify, “Few high-quality, prospective studies using frequent, serial PCR or antigen testing covering a broad range of occupations having been conducted, in part due to the cost and coordination required.” (Page 4, Lines 20-22).

Lines 36-38: I would call out occupational safety guideline creation.

Responses to reviewer: We have added “the development of occupational safety guidelines and regulations,” such that the sentence now reads, “Accurate data on the occupational risks of COVID-19 and other respiratory infections are essential for informing the development of occupational safety guidelines and regulations, transmission control measures and resource allocation (testing, personal protective equipment (PPE), etc.) (Page 4, Lines 26-29).”

PAGE 6

Line 6: machine translatable using which software?

Responses to reviewer: We have clarified that Google Translate was used to machine translate non-English articles (Page 5, Lines 48).

Line 38: I'm unfamiliar with this probability classification cut-off. Will the reader understand without additional information?

Responses to reviewer: There is not standard cut-off for manually verifying results from the National Institute for Occupational Safety & Health (NIOSH) Industry and Occupation Computerized Coding System (NIOCCS). The probability reported refers to the probability of correct classification to the six-digit level, however, we used coding to the two- and three-digit level in our review. With hierarchical coding systems such as the Standard Occupational Classification, manual or automated coding accuracy and agreement will be greater at the higher more aggregate levels. We manually verified a subset of records and based on i) the observation that most codes with a probability of correct classification of ≥ 0.8 to the six-digit level were correctly coded at the two- and three-digit level, which we used in our main analyses and ii) the number of records that could feasibly be verified manually. We now clarify in the main text that, “Coding was manually verified if there was insufficient information for classification or the probability of correct classification to the six-digit level was < 0.8 based on our review of a subset of the NIOCCS coded data (see supplement)” (Page 5, Lines 62-64) and have added further detail to the supplementary material.

PAGE 8

Line 20-23: I think there are additional issues beyond testing access, e.g., test characteristics, the proportion of asymptomatic infection.

Responses to reviewer: We now mention asymptomatic infection and test characteristics, “Seroprevalence studies may estimate the cumulative incidence of infection more accurately than diagnostic testing studies when access to testing and test performance are poor, and also can identify asymptomatic infections” (Page 5, Lines 59-62).

Line 34-39: I'm not following the 4% statistic--what is a mean risk of a positive test? Are you referring to percent positivity or a measurement of cumulative incidence at some point in the pandemic?

Responses to reviewer: We have now clarified that 4% refers to the cumulative incidence “The UK Office for National Statistics reported a slightly lower cumulative incidence of positive diagnostic or rapid tests for COVID-19 across 25 occupational groups of 4% (mean).” (Page 7, Lines 115-116).

PAGE 9

Lines 6-8: "to inform compliance with workplace safety regulations" is not really the domain of public health. I think the point here is that we need occupational risk data to inform understanding of transmission, occupational health and safety agency guidelines. And, on the public health side, we need to use these data to inform PPE and vaccine allocation.

Response to reviewer: We appreciate the reviewer's insight and comment and have revised this section of the discussion as suggestion. We have revised the sentence as per the reviewer's suggestion, “Accurate data on the occupational risks of respiratory infections, including SARS-CoV-2 are needed to inform understanding of transmission, occupational health and safety agency guidelines and allocation of resources (e.g., personal protective equipment and vaccines) during outbreaks and pandemics.” (Page 8, Lines 127-130).”

Lines 15-19: hmmm. Public health has very real constraints and must prioritize work especially during an early pandemic. I'm not sure this is a realistic recommendation. I'm also not totally clear on why this is being recommended?

Response to reviewer: We appreciate the reviewer's critique and have clarified the reasoning behind our recommendation, “As such, future population-based studies on respiratory infections should collect data on occupation. In the case of epidemic infection, collaboration between academic centres with the capacity to conduct large-scale studies and government agencies with expertise in disease surveillance and access to workplace data (e.g., public health, occupational health and safety) may be beneficial.¹⁰ Other authors have suggested the utility of occupational surveillance systems.¹⁵ However, the routine completion of the occupation field in electronic health records would also serve this purpose as well as informing patient reported outcome measures.” (Page 8, Lines 131-135).

Lines 35-36: contributing to misclassification?

Response to reviewer: We now clarify that, “Detailed descriptions of occupations were often lacking, potentially contributing to coding errors and misclassification.” (Page 8, Line 141).

PAGE 10:

Line 6: pandemic preparedness or response?

Response to reviewer: We intend to include both, because information on the occupational risks of infection collected during previous pandemics may inform regulations and policy in preparation of future pandemics (e.g., to facilitate a more rapid response and targeted dissemination of limited resources) and will also be useful for informing the initial response before new pandemic data become widely available. We now indicate that, “Carefully-designed, adequately powered seroprevalence studies with coverage of a broad range of occupations could improve our understanding of the occupational risk of SARS-CoV-2 and other respiratory infections and should be considered an element of pandemic preparedness and response.” **(Page 9, Line 162).**

Overall comments:

Why no traditional Figure 1?

Response to reviewer: We originally submitted the manuscript as a brief research article, which only allowed one figure to be included in the main text. Now we have added a PRISMA flow diagram to the supplementary material.

Very limited number of references, strongly suggest expanding literature review to inform manuscript.

Response to reviewer: We originally submitted the manuscript as a brief research article, which only allowed for ten references, but have expanded this. References for included studies were included in the supplementary material.

PRISMA check list?

Response to reviewer: We have added a PRISMA checklist to the supplementary material.

Reviewer: 2

Dr. Chao Wu, Nanjing University

Comments to the Author:

Boucher et al described and analyzed studies of SARS-CoV-2 seroprevalence by occupation prior to the widespread vaccine rollout, which seroprevalence studies covering a broad range of occupations were published in the first year of the pandemic. Results suggest considerable differences in seroprevalence between occupations, only few large, high-quality studies were done. Well designed studies are required to improve our understanding of the occupational risk of SARSCoV-2 and should be considered as an element of pandemic preparedness for future respiratory pathogens. However, after a careful reading of the article, I think that the importance of this review is not reflected, and the important conclusions of this study are not well presented.

The major issue are present study only described a broad range of occupations of seroprevalence in first years of pandemic, The epidemic situation, prevention and control policies, economy and working methods, as well as the behavior and habits of the people in each region were not analyzed and elaborated. Because the disease that spreads in a respiratory tract is in epidemic process, individual life habit and protection consciousness, economic status is the most important factor that causes epidemic, the characteristic of occupation is not obvious. Therefore, the description and analysis of the medical service industry, that is, the relevant research on these places of occupational exposure risk is more important.

Response to reviewer: We appreciate the reviewer’s comments and agree that individual habits, including the correct use of personal protective equipment and characteristics of occupations (e.g., if work is performed in a well-ventilated room) are important and probably mediate the occupational risks of contracting COVID-19 infection, as do other social determinants of health.[1] The role occupation recording as an important socioeconomic determinant and its use to define social class is acknowledged by Moscrop et al [2], they state, “the lack of socioeconomic data for individual patients has limited our understanding of the pandemic. In the UK, thanks to the inclusion of ‘occupation’ on death certificates, we know that security guards, taxi and bus drivers have had an especially high covid death rate.” However, a previous study of severe COVID-19 found that occupation is associated with inherent risks of infection over and above multiple socioeconomic (i.e., deprivation, educational attainment), work related (i.e., shift work, manual work, job tenure, working hours), health and lifestyle factors.[3] Consequently, our objective in conducting this review was to describe and synthesize studies reporting SARS-CoV-2 seroprevalence by occupation prior to the widespread roll-out of vaccines to primarily inform the development of occupational safety guidelines and regulations.

VERSION 2 – REVIEW

REVIEWER	Melissa Sutton Oregon Health Authority, Acute & Communicable Disease Prevention
REVIEW RETURNED	18-Nov-2022
GENERAL COMMENTS	Much improved. The intent of the manuscript is now well articulated. Congratulations to the authors. Line 21: suggest replace 'PCR' with 'molecular' or 'NAAT' to be more accurate